# A Machine Learning Approach to Duality in Statistical Physics

**Prateek Gupta** [1]  **Andrea E. V. Ferrari** [2][3]  **Nabil Iqbal** [4][5]

## Abstract

The notion of *duality* – that a given physical system can have two different mathematical descriptions – is a key idea in modern theoretical physics. Establishing a duality in lattice statistical mechanics models requires the construction of a dual Hamiltonian and a map from the original to the dual observables. By using neural networks to parameterize these maps and introducing a loss function that penalises the difference between correlation functions in original and dual models, we formulate the process of duality discovery as an optimization problem. We numerically solve this problem and show that our framework can rediscover the celebrated Kramers-Wannier duality for the 2d Ising model, numerically reconstructing the known mapping of temperatures. We further investigate the 2d Ising model deformed by a plaquette coupling and find families of "approximate duals". We discuss future directions and prospects for discovering new dualities within this framework. [1].

## 1. Background

A key concept in physics is *duality*, i.e. the idea that the same physical system can have two different mathematical descriptions. Duality sits at the heart of modern theoretical physics. In this work we seek to formalize the notion of duality in statistical physics in a manner that allows modern machine learning techniques to be used to systematically search for dualities.

---
[*]Equal contribution  [1]Max Planck Institute for Human Development, Berlin, Germany  [2]Deutsches Elektronen-Synchrotron DESY, Germany  [3]School of Mathematics, The University of Edinburgh  [4]Department of Mathematical Sciences, Durham University, UK  [5]Amsterdam Machine Learning Laboratory, University of Amsterdam, Netherlands. Correspondence to: Prateek Gupta <contact@pgupta.info>.

*Proceedings of the $42^{nd}$ International Conference on Machine Learning*, Vancouver, Canada. PMLR 267, 2025. Copyright 2025 by the author(s).

[1]Our implementation is publicly available at https://github.com/pg2455/physics_duality

**Background on duality:** Consider a statistical physics model with microstates $\sigma$ and Hamiltonian functional $H[\beta, \sigma]$, where $\beta$ are macroparameters such as the temperature. The model is determined by its partition function $Z = \sum_\sigma e^{-H[\beta,\sigma]}$. However, in nature we often have access to sets of *expectation values of observables* $O_\alpha(\sigma)$ (some real-valued functions of the microstates, e.g. correlation functions, with $\alpha$ being an arbitrary label)

$$\langle O_\alpha(\sigma) \rangle_H = \frac{1}{Z} \sum_\sigma O_\alpha(\sigma) \exp(-H[\beta, \sigma]). \quad (1)$$

It is a profound physical fact that occasionally there are alternative representations of these sets of correlation functions (see e.g. (Savit, 1980; Kramers & Wannier, 1941b;a; Peskin, 1978; Dasgupta & Halperin, 1981; Coleman, 1975; Wegner, 1971) for influential examples). That is, there exists another set of microstates $\tilde{\sigma}$, another Hamiltonian $\tilde{H}[\tilde{\beta}, \tilde{\sigma}]$ and for each observable $O_\alpha(\sigma)$ a dual observable $\tilde{O}_\alpha(\tilde{\sigma}_i)$ such that

$$\langle O_\alpha(\sigma) \rangle_H = \langle \tilde{O}_\alpha(\tilde{\sigma}) \rangle_{\tilde{H}}. \quad (2)$$

When this happens, we have a *duality* –the same physical system has at least two distinct mathematical descriptions, which may be useful for different reasons.

A prototypical example of such a duality is Kramers-Wannier duality for the 2d Ising model (Kramers & Wannier, 1941b). The 2d Ising model[2] consists of spins $\sigma_i = \pm 1$ living on the sites of a square lattice at temperature $\beta^{-1}$, with Hamiltonian that sums over neighbouring spins $\langle ij \rangle$

$$H[\beta, \sigma] = -\beta \sum_{\langle ij \rangle} \sigma_i \sigma_j. \quad (3)$$

It is a remarkable fact that the model described by (3) is precisely equivalent to a different 2d Ising model model with spins $\tilde{\sigma}_i = \pm 1$ living on the *dual* lattice, with a dual Hamiltonian of the same functional form

$$\tilde{H}[\tilde{\beta}, \tilde{\sigma}] = -\tilde{\beta} \sum_{\langle ij \rangle} \tilde{\sigma}_i \tilde{\sigma}_j \quad (4)$$

but with $\tilde{\beta}$ satisfying

$$\sinh(2\beta) \sinh(2\tilde{\beta}) = 1 . \quad (5)$$

---
[2]See e.g. (Kardar, 2007) for a textbook treatment.

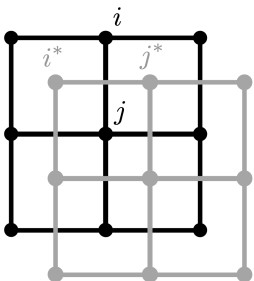

Figure 1. The two-point product of spins in the original frame $\sigma_i \sigma_j$ is related to the product of spins $\tilde{\sigma}_{i*} \tilde{\sigma}_{j*}$ in the dual frame, where $i^*, j^*$ are related to $ij$ as shown.

Note that this maps low temperatures to high temperatures. The fact that the functional form of the Hamiltonian is the same is exceptional, and in this case one can call the duality a *self*-duality.

Importantly, all observables constructed from the $\sigma_i$ can be mapped to observables of the $\tilde{\sigma}_i$. Consider for instance two neighbouring spins $\sigma_i$ and $\sigma_j$. We can build an observable $O_{ij} = \sigma_i \sigma_j$, which we call a *link product*. Then the KW duality implies that

$$\langle O_{ij} \cdots \rangle_H = \langle \tilde{O}_{ij}(\tilde{\sigma}) \cdots \rangle_{\tilde{H}}, \qquad \tilde{O}_{ij}(\tilde{\sigma}) = e^{-2\tilde{\beta}\tilde{\sigma}_{i*}\tilde{\sigma}_{j*}} \tag{6}$$

where the notation $\tilde{\sigma}_{i*}$ refers to sites on the dual lattice such that the link connecting sites $i^*$ and $j^*$ intersects the link connecting $i$ and $j$, as shown in Figure 1. The $\cdots$ indicate that this is an operator equation which holds for arbitrary insertions of operators and thus can be used to construct any expectation value of an even number of the $\sigma_i$. Appropriate products of the link products determine all correlation functions.[3]

Deformations of this prototypical duality have explicitly been studied in some special cases (Strycharski & Koza, 2013; Cobanera et al., 2011; Aasen et al., 2016). However, even in this controlled setup a lot remains to be understood and a fully systematic approach is not available. For instance, to our knowledge an explicit study of dual models to the plaquette model

$$H[\beta, \sigma] = -\beta \sum_{\langle ij \rangle} \sigma_i \sigma_j - \kappa \sum_{\langle ijkl \rangle} \sigma_i \sigma_j \sigma_k \sigma_l, \tag{7}$$

---

[3]Due to a $\mathbb{Z}_2$ symmetry the expectation value of a moment of an odd number of spins *formally* vanishes in a finite model, though as usual in the symmetry spontaneously broken phase this might not be observed in a simulation that uses local updates. We also note that the relation is modified if we consider precisely the same two-spin operator *twice*, i.e. $(\sigma_i \sigma_j)^2 = 1$, when a careful derivation of the duality shows that the right-hand side must be modified and is also identically 1.

where the second sum is a sum over plaquettes (squares) $\langle ijkl \rangle$, has not yet been performed.

In this work we tackle the problem of finding statistical physics dualities using machine learning. In particular, starting from an original model $(H, O)$ determined by a Hamiltonian $H$ and selected observables $O$, we formulate an optimization problem whose solution can recover *both* this model as well as dual descriptions $(\tilde{H}, \tilde{O})$. As a first step, we will focus on the 2d Ising model with Hamiltonian (3) and observable $O_{ij} = \sigma_i \sigma_j$, as well as on the plaquette model (7). We demonstrate that the optimization problem recovers the known dual to the 2d Ising model, thus offering an automated discovery of a duality. We furthermore give evidence for the absence of certain self-dualities of the plaquette model. As we shall see, also this negative result highlights interesting physical features.

**Previous work:** The problem of learning the parameters in a Hamiltonian from data is precisely that of training a Boltzmann machine, and has a very long history. Our case differs from the classical situation in that we are simultaneously learning a mapping of observables.

Other work on using machine learning to probe dualities in statistical physics includes (Betzler & Krippendorf, 2020). Section 3 of that work has some overlap with ours, where the key differences are: (1) in that work the input into the duality mapping is spin configurations sampled in the original frame, after which a second step of sampling is done: this is not exactly the usual setup for duality in physics, where one usually just samples once (importantly, in a dual frame) and then performs a deterministic mapping, as in our work. (2) It seems that the loss function used in that work cannot be formulated unless the duality mapping of temperatures is known already, and thus that this work cannot be used to find new dualities, which our formalism allows.

We also note work in the context of duality in quantum field theory (Bao et al., 2020).

## 2. Methodology

We now explain how, starting from the Hamiltonian $H[\beta, \sigma]$ of some statistical model on a lattice, we can learn candidates $\tilde{H}[\tilde{\beta}, \tilde{\sigma}]$ for dual models as well as a dictionary between original and dual observables. This includes learning the fact that the dual model is defined on a different lattice, such as the dual lattice.

**Framework and loss function:** We assume that $\tilde{H}$ can be written in terms of local couplings of spins:

$$\tilde{H}[\tilde{\beta}, \tilde{\sigma}_i] = -\tilde{\beta} \sum_{\langle ij \rangle} \tilde{\sigma}_i \tilde{\sigma}_j - \tilde{\kappa} \sum_{\langle ijkl \rangle} \tilde{\sigma}_i \tilde{\sigma}_j \tilde{\sigma}_k \tilde{\sigma}_l - \cdots \tag{8}$$

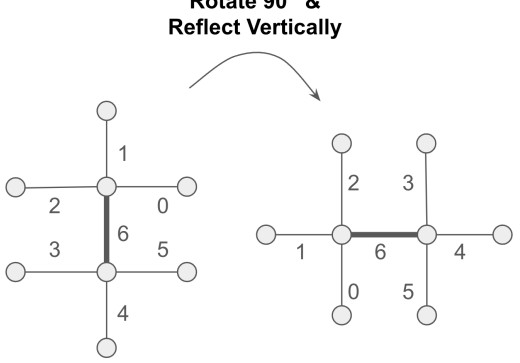

**Rotate 90° &
Reflect Vertically**

*Figure 2.* We parametrize $G$ as a neural network that takes neighboring links of a given link (in this case # 6) as its input. The assignment on horizontal links is related to that on vertical ones by a rotation and reflection.

where the couplings $(\tilde{\beta}, \tilde{\kappa}, etc.)$ are parameters to be learned. We would like to find dual representations of the link products $O_{ij}$ we described for the Ising model. We assume that the link product in the original model is mapped to *some* functions of *nearby* link products in the dual model, more precisely

$$\tilde{O}_{ij}(\tilde{\sigma}) = G(\{\tilde{\sigma}_k \tilde{\sigma}_l\}) \qquad (9)$$

where $\{\tilde{\sigma}_k \tilde{\sigma}_l\}$ is a set of link products such as the one shown in Figure 2.

$G$ is designed to be sufficiently flexible to recover models on lattices related in various ways to the original one. Note that a choice must be made about how to relate the assignment of link products neighbouring a horizontal link to the assignment of link products neighbouring a vertical link, as multiple choices are consistent with rotational invariance. In Figure 2 we display the choice used, which relates them by a rotation composed with a reflection. As we will see later, this choice is important for recovering the geometry of the dual lattice.

We now construct a loss function $\mathcal{L}$ that is minimized when all correlation functions of $O_{ij}$ and $\tilde{O}_{ij}$ agree on the two sides of the duality. This is similar to the matching of moments of two distributions, which is a standard problem, and for which one can construct general kernels that are minimized only when all of the moments of two distributions agree (see e.g. (Li et al., 2015)). Unfortunately, in the present case we cannot use kernels because of one conceptual and one technical problem: 1) certain moments need not be matched, as per Footnote 2, and 2) no notion of locality is embedded in standard moment matching. In the present case, correlation functions of faraway spins carry little information, and thus attempting to match their moments is a waste of computation.

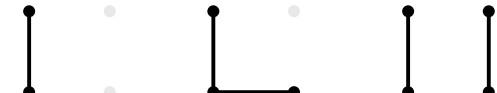

*Figure 3.* Examples of three features showing link products considered.

Instead we explicitly match *features* – i.e. moments of a small number of nearby link products, as shown in Figure 3 – which we then spatially average over the lattice. Denoting these features as $\phi^a$ with $a$ running over features, we then construct the loss

$$\mathcal{L}(G, \tilde{H}) = \sum_a \ell^a \ell^a \qquad \ell^a = \langle \phi^a[G(\tilde{\sigma}_i)] \rangle_{\tilde{H}} - \langle \phi^a[\sigma_i] \rangle_H$$
(10)

$\ell^a$ can be thought of as a vector in feature space indicating how far apart the two theories are.

For the 2d Ising model, it is clear that this loss can be minimized in two scenarios: (a) $\tilde{H} = H$ and $G(\tilde{\sigma}_i \tilde{\sigma}_j) = \tilde{\sigma}_i \tilde{\sigma}_j$, i.e., the original model is rediscovered, or (b) $\tilde{H} \neq H$ and $G(\tilde{\sigma}_i \tilde{\sigma}_j) \neq \tilde{\sigma}_i \tilde{\sigma}_j$, representing a nontrivial dual model where (selected) moments nevertheless perfectly match those of the original model. The plaquette model has no dual that is known explicitly.

**Optimization:** We now need to solve the following optimization problem:

$$G^*, \tilde{H}^* = \arg\min_{G, \tilde{H}} \mathcal{L}(G, \tilde{H}) \qquad (11)$$

$G$ is represented by a neural network with parameters $\theta$, $G = G_\theta$.

Algorithm 1 outlines the procedure for optimization. Given a trial set of parameters $\theta$ and couplings for the dual Hamiltonian $\tilde{\beta}_a$, we simultaneously perform Markov Chain Monte Carlo (MCMC) sampling from the original and dual Hamiltonians using a standard Metropolis algorithm to obtain spin configurations $\sigma_i$ and $\tilde{\sigma}_i$ drawn from the appropriate distributions respectively. We can then evaluate the expectation values in (10), and compute the loss $\mathcal{L}$.

To minimize it we also need to compute gradients $\partial_\theta \mathcal{L}$ and $\partial_{\tilde{\beta}_a} \mathcal{L}$. For $\theta$ this can be done straightforwardly using conventional automatic differentiation techniques. For the $\tilde{\beta}_a$ we cannot backpropagate through a stochastic sampler, but explicit differentiation shows that we can relate the gradients to expectation values that can be evaluated through MCMC sampling from the dual Hamiltonian. For concreteness we demonstrate the argument with only a single nonzero coupling $\tilde{\beta}$ in (8), but the generalization to other couplings (and in particular the plaquette coupling $\tilde{\kappa}$) is immediate. For any

function of spins $\mathcal{O}[\tilde{\sigma}]$ we have

$$\langle \mathcal{O} \rangle_{\tilde{H}} \equiv \frac{1}{Z(\tilde{\beta})} \sum_{\{\tilde{\sigma}_i\}} \mathcal{O}[\tilde{\sigma}] e^{\left(\tilde{\beta} \sum_{\langle ij \rangle} \tilde{\sigma}_i \tilde{\sigma}_j\right)} \tag{12}$$

where $Z(\tilde{\beta}) \equiv \sum_{\{\tilde{\sigma}_i\}} e^{\left(\tilde{\beta} \sum_{\langle ij \rangle} \tilde{\sigma}_i \tilde{\sigma}_j\right)}$ and the sum over $\{\sigma_i\}$ runs over all spin configurations. Now we have

$$\partial_{\tilde{\beta}} \mathcal{L} = 2 \sum_a \ell^a \partial_{\tilde{\beta}} \langle \phi^a[G(\tilde{\sigma}_i)] \rangle_{\tilde{H}}, \tag{13}$$

where we have used the definition of $\ell^a$ in (10). From (12) the gradient of any observable with respect to $\tilde{\beta}$ is

$$\partial_{\tilde{\beta}} \langle \mathcal{O} \rangle_{\tilde{H}} = -\langle \mathcal{O} \rangle_{\tilde{H}} \langle \sum_{\langle ij \rangle} \tilde{\sigma}_i \tilde{\sigma}_j \rangle_{\tilde{H}} + \sum_{\langle ij \rangle} \langle \tilde{\sigma}_i \tilde{\sigma}_j \mathcal{O} \rangle_{\tilde{H}} \tag{14}$$

where the first term comes from differentiating $Z(\tilde{\beta})$ and the second from differentiating inside the Boltzmann measure weighting each configuration in (12). Using this expression to evaluate (14) for $\mathcal{O} = \phi^a[G(\tilde{\sigma}_i)]$ we find:

$$\partial_{\tilde{\beta}} \mathcal{L} = -2 \sum_a \ell^a \left\langle \left( \langle \sum_{\langle ij \rangle} \tilde{\sigma}_i \tilde{\sigma}_j \rangle_{\tilde{H}} - \sum_{\langle ij \rangle} \tilde{\sigma}_i \tilde{\sigma}_j \right) \phi^a[G_\theta(\tilde{\sigma})] \right\rangle_{\tilde{H}} \tag{15}$$

We can now evaluate the expectation value by MCMC sampling from the dual Hamiltonian. We note that this evaluation is computationally expensive, as each gradient step requires us to equilibrate an MCMC chain. For training conventional Boltzmann machines one can use more efficient approaches such as contrastive divergence (Carreira-Perpinan & Hinton, 2005). Due to the presence of the mapping $G$, we are not aware of a similarly efficient algorithm in our case, and indeed all likelihood-based approaches seem conceptually difficult.

---

**Algorithm 1** Machine learning for finding statistical mechanical duality

---

1: **Inputs:** $\beta$, $\eta$ (learning rate), $N$ (number of samples)
2: **Initialize:** $\tilde{\beta}_0 \in \mathbb{R}$, $\theta \in \mathbb{R}^d$
3: **for** each epoch $t = 1, 2, \ldots, T$ **do**
4:     Draw $N$ samples $\{\sigma_i\}_{i=1}^N \sim p(\sigma|\beta)$
5:     Draw $N$ samples $\{\tilde{\sigma}_i\}_{i=1}^N \sim p(\tilde{\sigma}|\tilde{\beta})$ where $\tilde{\beta} \neq \beta$
6:     Compute the loss $\mathcal{L} = \frac{1}{N} \sum_{i=1}^N \mathcal{L}(\sigma_i, G_\theta(\tilde{\sigma}_i))$
7:     Compute the gradients $\partial_{\tilde{\beta}} \mathcal{L}$ and $\partial_\theta \mathcal{L}$
8:     Update the parameters:

$$\tilde{\beta}_{t+1} \leftarrow \tilde{\beta}_t - \eta \partial_{\tilde{\beta}} \mathcal{L}$$
$$\theta_{t+1} \leftarrow \theta_t - \eta \partial_\theta \mathcal{L}$$

9:     **if** $\mathcal{L}$ has not improved for the last $X$ epochs **then**
10:         **Stop the optimization**
11:     **end if**
12: **end for**

---

**Improving convergence through variance reduction in gradient estimation.** Theoretically, computing gradients as described above should be sufficient. In practice, we observe significant noise, which hinders the optimization process. Computing gradients using MCMC inherently has high variance, making the optimization procedure highly sensitive to inefficient sampling. We find this problem to be especially severe in the case of two or more couplings.

To address this, we leverage the fact that in our training procedure, the target system remains unchanged across steps. This allows us to aggregate the target feature vector over multiple steps, thereby stabilizing it over time. Let $t^a = \langle \phi^a[\sigma_i] \rangle_H$ denote the running expectation of the target feature after a sufficient number of steps, such that its variance is minimized. The difference in feature vectors can be written as

$$\ell^a = \langle \phi^a[G(\tilde{\sigma}_i)] \rangle_{\tilde{H}} - t^a \tag{16}$$

For the second issue, we mitigate the variance in MCMC-based gradient estimation using control variates, a common variance reduction technique. Practical constraints limit our ability to obtain sufficiently large samples that accurately reflect the underlying distribution. To counter this, we introduce a constant baseline as a control variate (Mohamed et al., 2020; Greensmith et al., 2004). Since the target stabilizes over time, our baseline includes the target feature itself, effectively reducing variance in the gradient estimation process. Thus, gradients are estimated as follows:

$$\partial_{\tilde{\beta}} \mathcal{L} = -2 \sum_a \ell^a \left\langle \left( \sum_{\langle ij \rangle} \langle \tilde{\sigma}_i \tilde{\sigma}_j \rangle_{\tilde{H}} - \sum_{\langle ij \rangle} \tilde{\sigma}_i \tilde{\sigma}_j \right) \times \right.$$
$$\left. (\phi^a[G_\theta(\tilde{\sigma})] - t^a) \right\rangle_{\tilde{H}}. \tag{17}$$

Note that the extra term here relative to (14) is that in $t^a$; as usual for such baselines, it is proportional to $\partial_{\tilde{\beta}} \log p$ and thus vanishes in expectation, but reduces the variance.

## 3. Experiments

In this section, we describe some simple experiments using the above machinery.

**Neural Network architecture for $G$:** For a given link product in the dual frame we assemble the 7 nearby links shown in Fig. 2 into a 7-dimensional vector $\mathbf{f}_{\langle ij \rangle} \in (\mathbb{Z}_2)^7$, where each element of the vector is the product of the two spins living on the two ends of the link. We consider a simplistic neural network acting on this input, with parameters formed by $\theta_1 \in \mathbb{R}^7$, and scalars $\theta_2$ and $\theta_3$. We opt for hard attention using Gumbel-Softmax (Jang et al., 2016) so

that only a few of the seven nearby links are utilized in the prediction task. Thus, the mapping is defined by,

$$G_\theta(\mathbf{f}_{\langle ij \rangle}) = \theta_2 \cdot \text{Gumbel-Softmax}(\theta_1)^T \mathbf{f}_{\langle ij \rangle} + \theta_3 \quad (18)$$

As the elements of $\mathbf{f}_{\langle ij \rangle}$ are $\pm 1$, a very simple network provides a very expressive function. In our experiments, we initialize $\theta_2$ and $\theta_3$ from a uniform distribution, $\mathcal{U}(-1, 1)$, and $\theta_1$ from a normal distribution, $\mathcal{N}(0, 1)$.

### 3.1. Original 2d Ising model

We take our original Hamiltonian $H$ to be that of the 2d Ising model (3), and we take the dual Hamiltonian $\tilde{H}$ in (8) to have only one non-zero parameter $\tilde{\beta}$ (and so $\tilde{\kappa} = 0$, etc.).

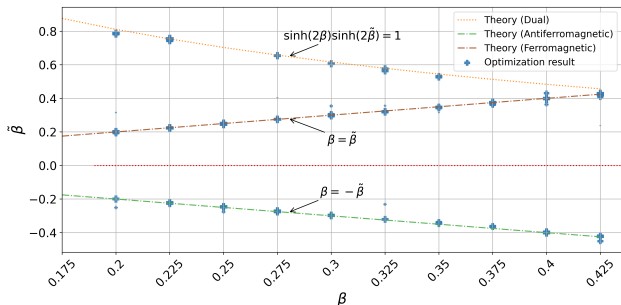

Figure 4. Final $\tilde{\beta}$ as found by the deep learning framework closely matches that of the theoretical results. Points are scaled by the negative logarithm of the best loss such that the size of the points is inversely proportional to the loss. We cap the minimum size so that smaller points are visible. The loss is a minimum along two fronts, i.e, original frame $\beta = \pm\tilde{\beta}$ and the dual frame along the lines $\sinh(2\beta)\sinh(2\tilde{\beta}) = 1$.

.

**Rediscovery of the 2d Ising duality.** In Figure 4, we show the result of deploying the above machinery on different model values of $\beta$ on an $8 \times 8$ lattice with periodic boundary conditions. For each value of the input $\beta$, we ran a total of 10 optimizations, five from each of the two initializations of $\tilde{\beta}$, i.e., $\tilde{\beta}_0 = 0.2$ and $\tilde{\beta}_0 = 0.5$. Due to the randomness involved in MCMC sampling, each seed is expected to be an independent run.

We record the value of $\tilde{\beta}$ obtained. There are three branches of solutions: the original model $\tilde{\beta} = \beta$, the dual model $\sinh(2\beta)\sinh(2\tilde{\beta}) = 1$, and an antiferromagnetic analogue of the original model $\tilde{\beta} = -\beta$. The latter is equivalent to the original frame, and is obtained by making the change of variables $\sigma_i \to -\sigma_i$ on every other site, thus flipping the sign of $\beta \to -\beta$. Note that the existence of the dual branch of solutions can be viewed as a numerical "rediscovery" of the KW duality line

$$\sinh(2\beta)\sinh(2\tilde{\beta}) = 1 \quad (19)$$

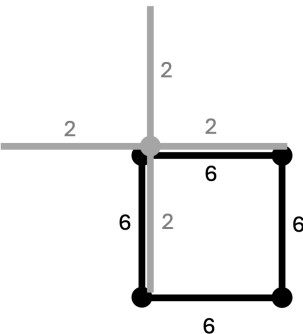

Figure 5. Emergence of dual lattice: e.g. if four original links (marked by 6) form a square, the corresponding four links that are referenced by the neighbour mapping (marked by 2) in Figure 2 form a cross, as expected for the dual lattice.

Interestingly, we find that the method does not perform reliably as we approach the phase transition $\beta = \beta_c \approx 0.44$, where the dual and original branches coincide. In addition, we find that it does not work equally well for $\beta > \beta_c$, when the original frame is in the symmetry-broken phase. We show the same plot for this phase in Supplementary Material. This is somewhat reminiscent of known difficulties in learning parameters of Hamiltonians at high $\beta$ (see e.g. Appendix B of (Haah et al., 2024)) and deserves further study.

Further details on the experiments (including an exploration on how they depend on the system size) are shown in the Supplementary Material.

It is interesting to ask how the model recovers the structure of the dual *lattice*, as well as the dual observables. The attention mechanism used encourages the model to use only a single link of the input, and for the runs that find the dual temperature this ends up using the links numbered either 2 or 5 instead of the original 6 in Figure 2. As we show in an example in Figure 5, this is equivalent to finding the dual lattice from the original. Note that here it is important that we relate horizontal to vertical links by the composition of a rotation *and* reflection as shown in Figure 2; other choices will not result in the possibility of finding the dual lattice, and indeed in our experiments they do not find a duality. The optimized values of $G_\theta$ closely match theoretical results $\tilde{O}_{ij}(\tilde{\sigma}) = e^{-2\tilde{\beta}\tilde{\sigma}_{i*}\tilde{\sigma}_{j*}}$, as shown in more details through the sampled training trajectories in the Supplementary Material.

In this approach, the one-to-one mapping of $\beta$ to $\tilde{\beta}$ is only found numerically; one could possibly supplement this numerical determination with symbolic regression (Schmidt & Lipson, 2009) to obtain an analytic formula such as (19), but in more complicated examples of the duality we do not expect there to necessarily exist a simple analytic formula and thus have not explored this.

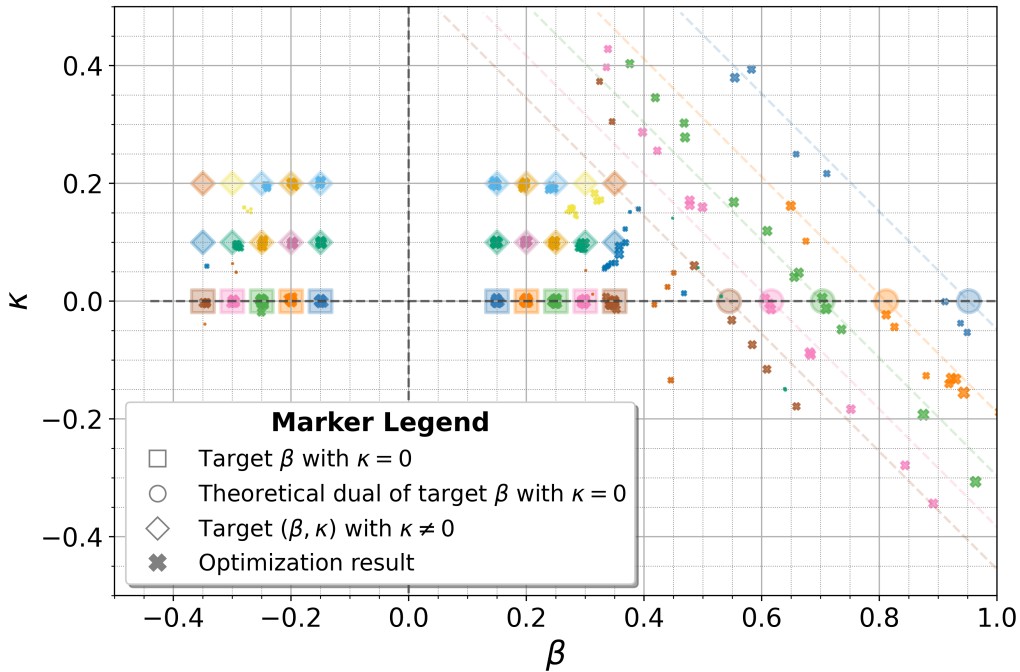

Figure 6. We display the both the target couplings $(\beta, \kappa)$ and the output couplings $(\tilde{\beta}, \tilde{\kappa})$ from the optimization procedure. We also indicate theoretical duals to models with $(\beta \neq 0, \kappa = 0)$. Note that duals to models with $\kappa \neq 0$ are not known. Similar to Figure 4, the size of the points is inversely proportional to the loss. The optimization often finds the original frame $(\beta, \kappa)$ (or its antiferromagnetic image $(-\beta, \kappa)$). When the target $\kappa = 0$, duals are still recovered, though accompanied by clusters along the lines $\tilde{\beta} + \tilde{\kappa} = \text{const}$, the reason for which we discuss in Section 3.2.

## 3.2. Plaquette 2d Ising model

We now turn to a slight generalization of the familiar Ising model by adding an extra 4-spin coupling:

$$H[\beta, \kappa; \sigma_i] = -\beta \sum_{\langle ij \rangle} \sigma_i \sigma_j - \kappa \sum_{(ijkl)} \sigma_i \sigma_j \sigma_k \sigma_l \qquad (20)$$

where in the second term we take the product of four spins around an elementary square plaquette. This "2d Ising plaquette model" is no longer exactly solvable and has been previously studied as a nontrivial testbed for ML approaches to statistical physics problems (see e.g. (Huang & Wang, 2017; Wang, 2017)). This model again has a disordered phase at small $(\beta, \kappa)$ and an ordered phase at larger couplings. For completeness we present a simple mean-field description of the phase diagram in Appendix A. As mentioned in the introduction, no precise Kramers-Wannier duality for this exact model is known for finite $\kappa$.

We now discuss the results from applying the machinery above to search for a duality with the same functional form, i.e.

$$H[\tilde{\beta}, \tilde{\kappa}; \tilde{\sigma}_i] = -\tilde{\beta} \sum_{\langle ij \rangle} \tilde{\sigma}_i \tilde{\sigma}_j - \tilde{\kappa} \sum_{(ijkl)} \tilde{\sigma}_i \tilde{\sigma}_j \tilde{\sigma}_k \tilde{\sigma}_l \qquad (21)$$

The results are shown in Figure 6. We ran a total of 300 ex-

periments, corresponding to 20 runs for each combination of $(\beta, \kappa)$ in the set $\{0.15, 0.2, 0.25, 0.3, 0.35\} \times \{0.0, 0.1, 0.2\}$. Here we see that the output from the optimization often recovers the original frame $(\beta, \kappa)$ (or, as above, its antiferromagnetic image $(-\beta, \kappa)$). However we do not find any new dual theories when $\kappa \neq 0$.

This is somewhat expected. Recall that away from the conventional 2d Ising model, one does not necessarily expect the dual Hamiltonian to take precisely the same functional form – i.e. have exactly the same nonzero couplings – as the original. Thus, our analysis gives evidence against such an hypothetical fortuitous scenario. A more systematic approach would require us to turn on a larger number of couplings (e.g. we could imagine allowing for all couplings that couple spins in a neighbourhood of a given size).

We should still however ask whether – now that *model* $\tilde{\kappa}$ is allowed to vary – we can still find the standard expected KW duals to theories with the *target* $\kappa = 0$, which are shown as circles in Figure 6. Though we find these, we also find clusters of theories along lines of the form $\tilde{\beta} + \tilde{\kappa} = \text{const}$ emerging from the known duals. To understand the physics behind this surprising fact, note that this happens when the dual model is deep in the ordered phase. Consider now a typical configuration of spins in this phase. To good approx-

imation, the spins will all be pointing in the same direction, i.e. we may imagine $\tilde{\sigma}_i = 1$ for almost all $i$, with occasional very rare spin flips to $\tilde{\sigma}_i = -1$. From the Hamiltonian (21) we can compute that the energy cost to flip a spin against this background is $8(\tilde{\beta} + \tilde{\kappa})$, so the probability to flip the spin (as compared to keeping it constant) behaves as $p \sim e^{-8(\tilde{\beta}+\tilde{\kappa})}$. This single spin flip probability – which depends only on the combination $\tilde{\beta} + \tilde{\kappa}$ – will determine essentially all of the observables, as the chance of flipping two nearby spins is itself even smaller. Thus we see that almost all observables depend only on $\tilde{\beta} + \tilde{\kappa}$, and the optimization algorithm finds it difficult to distinguish points along this line.

This "approximate duality" is not specific to this model and will essentially happen any time we are dealing with a dual frame which is deep in an ordered phase. It reflects the fact that all systems which can be described by a dilute gas approximation (i.e. described by a density of dilute objects such as rare flipped spins) have a kind of universality in that all observables are determined by a single parameter: the probability of the rare event, in our case $e^{-8(\tilde{\beta}+\kappa)}$. In this sense matching this parameter alone will result in a "dual" description. To localize the system along the line we need to increase the precision of our observables so that they can be sensitive to even lower probability events involving the interaction of multiple rare events. In practice this will likely require more sample-efficient optimization techniques.

**Empirical support for approximate duality.** We now show that a broad set of moments – including correlation length – is accurately matched across approximate duals. To do so in a completely generalizable fashion, (a) we evaluate moments that were not included in the training loss, (b) we compute these moments on $24 \times 24$ lattices, beyond the training regime, (c) we include approximate duals along the hypothesized line for comparison. Due to computational constraints, training directly on large lattices like $24 \times 24$ is infeasible. Instead, we apply learned mappings $G_\theta$ from $8 \times 8$ lattices to estimate features on larger systems without retraining.

Figure 7 shows that the average moments across all approximate duals for $\beta_0 \in \{0.2, 0.25, 0.3\}$ exactly match that of the theoretical dual frames. Due to the lack of space, we provide an extensive comparison in the supplementary material.

**Impact of variance reduction on convergence.** We compare our proposed algorithm, which incorporates variance reduction techniques for gradient estimation in (17), with the theoretically derived gradients from (15) in this section.

To quantify deviations, we define the $\Delta$ as the maximum absolute deviation in $\beta$, $G_\theta(+1)$, $G_\theta(-1)$, $\kappa$. Figure 8 presents the number of experiments where $\Delta$ remains below

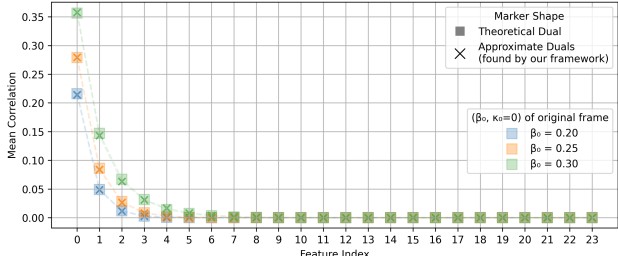

*Figure 7.* The moments formed from the product of consecutive links forming a linear chain in a lattice of size $24 \times 24$ match across approximate duals found from the framework across $\beta_0 \in \{0.2, 0.25, 0.3\}$ and their corresponding theoretical duals. We cover extensive comparisons on other types of moments in the supplementary material.

a given threshold $\epsilon$. Ideally, we aim to maximize the area under this curve.

We conduct 10 experiments for each combination of $(\beta, \kappa)$ in the set $\{0.25\} \times \{0.0, 0.1\}$, evaluating both conditions: with and without variance reduction. The dominance of blue lines (with variance reduction) over orange lines (no variance reduction) highlights that methods without variance reduction techniques exhibit poor convergence.

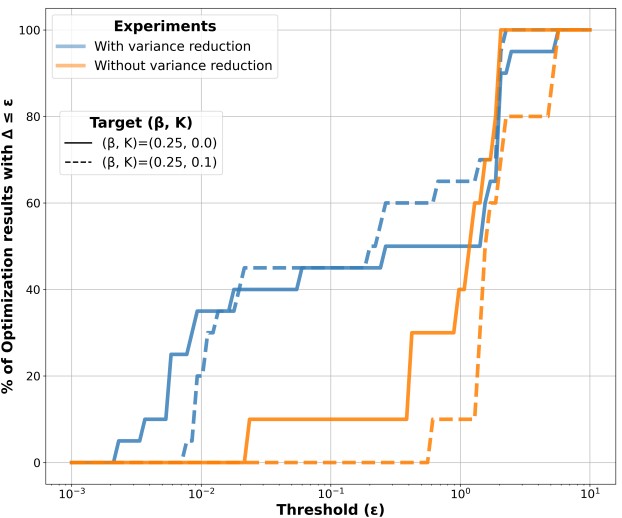

*Figure 8.* Here we benchmark the control variate technique that we use, determining how many of our runs recover a known target theory to within a given tolerance $\epsilon$. As the tolerance is relaxed more and more runs are counted; the area under this curve is a measure of the success of the algorithm. We see a clear increase in efficiency from using the variance reduction technique.

**Interpreting model learning behavior.** Figure 10 provides a visual representation of mappings learned by the models for $\sigma_i \sigma_j = \pm 1$, in the optimization results obtained from Figure 6. Notably, the majority of runs converge to

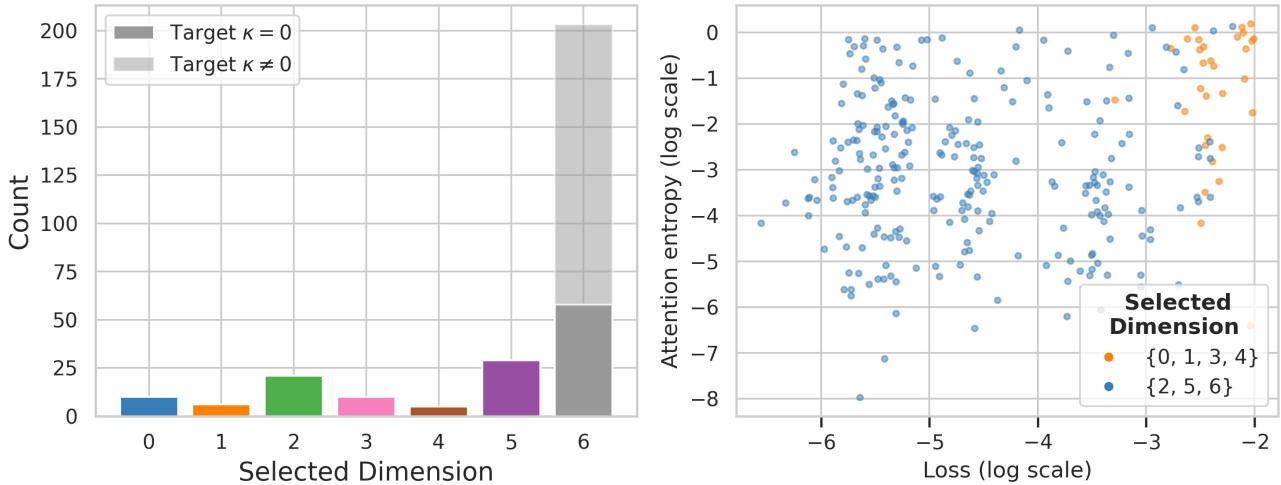

*Figure 9. Left.* The attention mechanism generally picks a single link to determine the observable. In the labeling of Figure 2, we display the links chosen at the endpoint of the run for cases where $\kappa = 0$ and where $\kappa \neq 0$. Note that when $\kappa = 0$ generally link 6 is obtained (which indicates the rediscovery of the original theory), and when $\kappa \neq 0$ there is a reasonable chance to find links 2 or 5, indicating a mapping to the dual lattice and Kramers-Wannier duality, as explained around Figure 5. *Right*, we demonstrate that runs where links $(2, 5, 6)$ are chosen generally have much lower loss and attention entropy.

the expected mappings, either learning the original mapping in the ferromagnetic phase with $G_\theta(x) = x$ or in the anti-ferromagnetic phase with $G_\theta(x) = -x$. The results that align with "approximate duality" exhibit mapping values close to those expected under perfect duality when $\kappa = 0$. This visualization gives a close look into what models are actually learning at an internal level.

To further investigate which mappings the models predominantly learned, we examine the frequency of selected links in the left panel of Figure 9. As discussed earlier, only links numbered 2 and 5 correspond to the correct dual mapping, while link 6 represents the original link. Our optimization results predominantly select links 2, 5 or 6, with the majority favoring the original link, particularly in cases where $\kappa \neq 0$. On the right panel of Figure 9, we plot, as a function of the loss, the entropy of link selection computed as $\sum_{l=0}^{6} \theta_{1l} * \log \theta_{1l}$, where $l$ corresponds to the links. Interestingly, as indicated by the spatial clustering of blue and orange dots, optimization results with lower uncertainty in link selection (i.e., lower entropy) exhibit a higher probability of selecting the correct link. These optimization runs also achieve lower loss values, reaching as low as $1e - 7$. Thus, a high entropy in link selection is associated with suboptimal optimization outcomes.

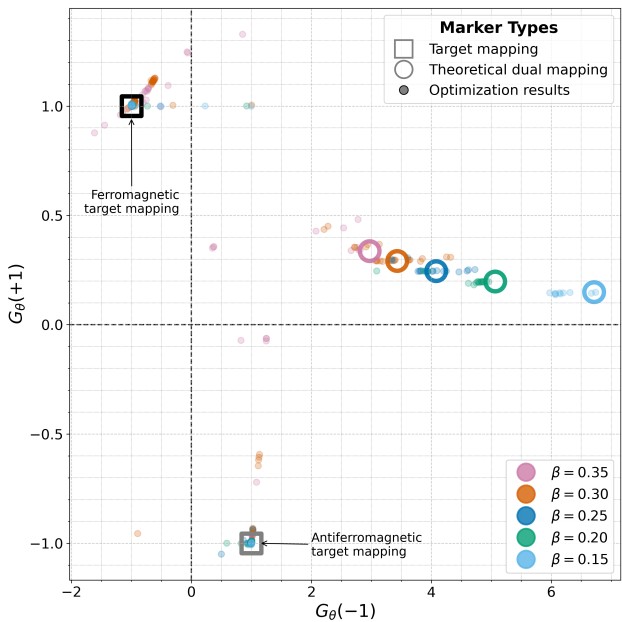

*Figure 10.* We display the mapping functions found by the algorithm, plotting $G_\theta(+1)$ against $G_\theta(-1)$ and indicating the trivial ferromagnetic solution (where $G_\theta(x) = x$), the antiferromagnetic solution (where $G_\theta(x) = -x$) and the non-trivial duality (where $G_\theta(x) = \exp(-2\tilde{\beta}x)$). The FM and AFM clusters contain many runs.

## 4. Conclusions

Above we have explained how the process of finding dualities can be automated, demonstrating the mechanism by "rediscovering" the well-known Kramers-Wannier duality of

the 2d Ising model, and by testing our system on the more general plaquette model. This is only a proof of principle, and much work remains to be done.

For example, as discussed in Section 2, at present we match a number of features which are constructed by hand. It would be ideal to find a kernel that allows matching of all the required moments while simultaneously giving lower weight to those involving faraway spins. On the operational side, it would be helpful to have a more efficient way of training; contrastive divergence fails here as there appears to be no simple way to map the likelihood of a single spin configuration across the duality.

On the physics side, we hope to use such techniques to find new dualities or to understand approximate dualities. One direction that we have initiated above is to search for Kramers-Wannier duals of deformed Ising models, where extra spin-spin couplings such as the plaquette term above have been added to the action. While some results exist for specific models (Strycharski & Koza, 2013; Cobanera et al., 2011; Aasen et al., 2016), we are not aware of a completely general approach that provides very explicit results. Our experiments show that adding more couplings generically increases the difficulty, highlighting the need for more sample-efficient techniques. Finally, a less concrete but far more exciting direction would be if one could use the approach to find entirely new dualities, unconnected to any existing ones. We hope to return to this in the future.

## Acknowledgments

We are very grateful to Roberto Bondesan, Arkya Chaterjee, Tarun Grover, Tyler Helmuth, Theo Jacobson, John McGreevy, Takuo Matsubara, Salvatore Pace and Tin Sulejmanpasic for helpful discussions. We thank the anonymous referees for their useful feedback. This work was supported by a grant from the Simons Foundation (PD-Pivot Fellow-00004147, NI). NI is supported in part by the STFC under grant number ST/T000708/1. AEVF was in part supported by the EPSRC Grant EP/W020939/1 "3d N=4 TQFTs". This work has made use of the Hamilton HPC Service of Durham University.

## Impact Statement

Our work holds significant potential to advance knowledge in statistical physics. At this stage, we do not anticipate any negative societal impacts.

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

## A. Mean-field discussion of plaquette model

Here we establish some basic features of the plaquette model defined by

$$H[\beta, \kappa; \sigma_i] = -\beta \sum_{\langle ij \rangle} \sigma_i \sigma_j - \kappa \sum_{(ijkl)} \sigma_i \sigma_j \sigma_k \sigma_l \tag{22}$$

There is no known exact solution to this model for all $\beta, \kappa$. On general grounds we expect a disordered phase at small $(\beta, \kappa)$ and an ordered phase for larger $(\beta, \kappa)$. We present a simple mean-field discussion of the Hamiltonian to confirm this expectation, noting that while we expect gross features of the phase diagram to survive, it is not expected to be quantitatively correct in $d = 2$.

Denote the true probability distribution for this model by

$$p_{\beta, \kappa}(\sigma_i) = \frac{1}{Z(\beta, \kappa)} \exp(-H[\beta, \kappa; \sigma_i]) \tag{23}$$

As usual we perform a mean-field treatment by postulating a simpler distribution $q_\phi(\sigma_i)$ labeled by some variational parameters $\phi_i$ and minimize the KL divergence between the true distribution and the variational one:

$$D_{KL}(q_\phi || p_{\beta, \kappa}) \equiv \left\langle \log \frac{q_\phi(\sigma_i)}{p_{\beta, \kappa}(\sigma_i)} \right\rangle_q = \langle H[\beta, \kappa; \sigma_i] + \log q_\phi(\sigma_i) \rangle_{q_\phi} + \text{const} \tag{24}$$

The constant contains the intractable partition function $Z(\beta, \kappa)$, but it is independent of the variational parameters and so can be neglected. In physics this is precisely the minimzation of the free energy $E - TS$, where our choice of where to place the factors of $\beta$ means that factors of $T$ appear slightly differently. We now pick a trial $q_\phi$ which is factorized on the sites, i.e.

$$q_\phi[\sigma_i] = \prod_i \frac{\exp(\phi_i \sigma_i)}{2 \cosh(\phi_i)} \tag{25}$$

where our variational parameter $\phi_i$ on each site can be thought of loosely as a classical coarse-grained field. This is the most general factorized distribution for a binary variable.

Computing the KL divergence for the choice where $\phi_i = \phi$ is constant on all sites we find

$$D_{KL}(q_\phi || p_{\beta, \kappa}) = -2\beta \tanh^2 \phi - \kappa \tanh^4 \phi + \phi \tanh \phi - \log(2 \cosh(\phi)) \tag{26}$$

This function always has a stationary point at $\phi = 0$ due to the $\mathbb{Z}_2$ symmetry $\phi \to -\phi$. Exploration of the minima indeed shows that this stationary point is a minimum of the free energy for small $(\beta, \kappa)$ but is no longer a minimum at large $(\beta, \kappa)$. The existence of a minimum of the free energy at a nonzero value of $\phi$ indicates an ordered phase with spontaneous breaking of the $\mathbb{Z}_2$ symmetry.

We focus on the limiting cases: at $\kappa = 0$ there is a second order transition at $\beta = \frac{1}{4}$ (this is the standard result for the mean-field treatment of the 2d Ising model), and at $\beta = 0$ there is a first-order transition at $\kappa \approx 0.688$, where the precise value was found numerically through balancing the free energy at the trivial and nontrivial minima of (26). Numerical exploration shows that the phase transition line connects these two points straightforwardly.

We include this discussion for completeness, noting that the quantitative features are unlikely to survive (e.g. note the well-appreciated fact that even at $\kappa = 0$ the true value for the transition at $\beta \approx 0.44$ differs significantly from the mean-field estimate $\beta_{MF} = 0.25$). For that reason we have not found the precise location of the change from second-order to first-order, as it is unlikely to be accurate for the real model. However the topology of the phase diagram is likely to have the shape shown, as is borne out by our numerical experiments.

## B. Neural network training

Our models are all implemented in PyTorch (Paszke et al., 2017). We used the Adam (Kingma & Ba, 2014) optimizer with the learning rate of 0.01. Moreover, we used the early stopping criterion to stop the training if the loss didn't improve over 200 epochs. We ran the sampler in each experiment to generate 1000 samples for the lattice. We ran the training for a maximum of 25000 epochs, and our runs took about 1-3 hours each. The experiments in the main paper are run on the lattice size of $8 \times 8$.

## C. Typical training curves

We provide some further details on our experimental results. The plots below are representative and were obtained with the control variate technique.

Figure 11 shows runs for $\beta = 0.25$ grouped by $\beta_0$ and frame discovered by the runs, illustrating how the training progresses under different scenarios. For the seeds where either the dual or original frame is recovered, the loss goes to 0. Further, we track the entropy of Gumbel-Softmax($\theta_1$) to assess how the algorithm is weighing each feature. A value of 0 corresponds to a strong preference for one out of the seven input links.

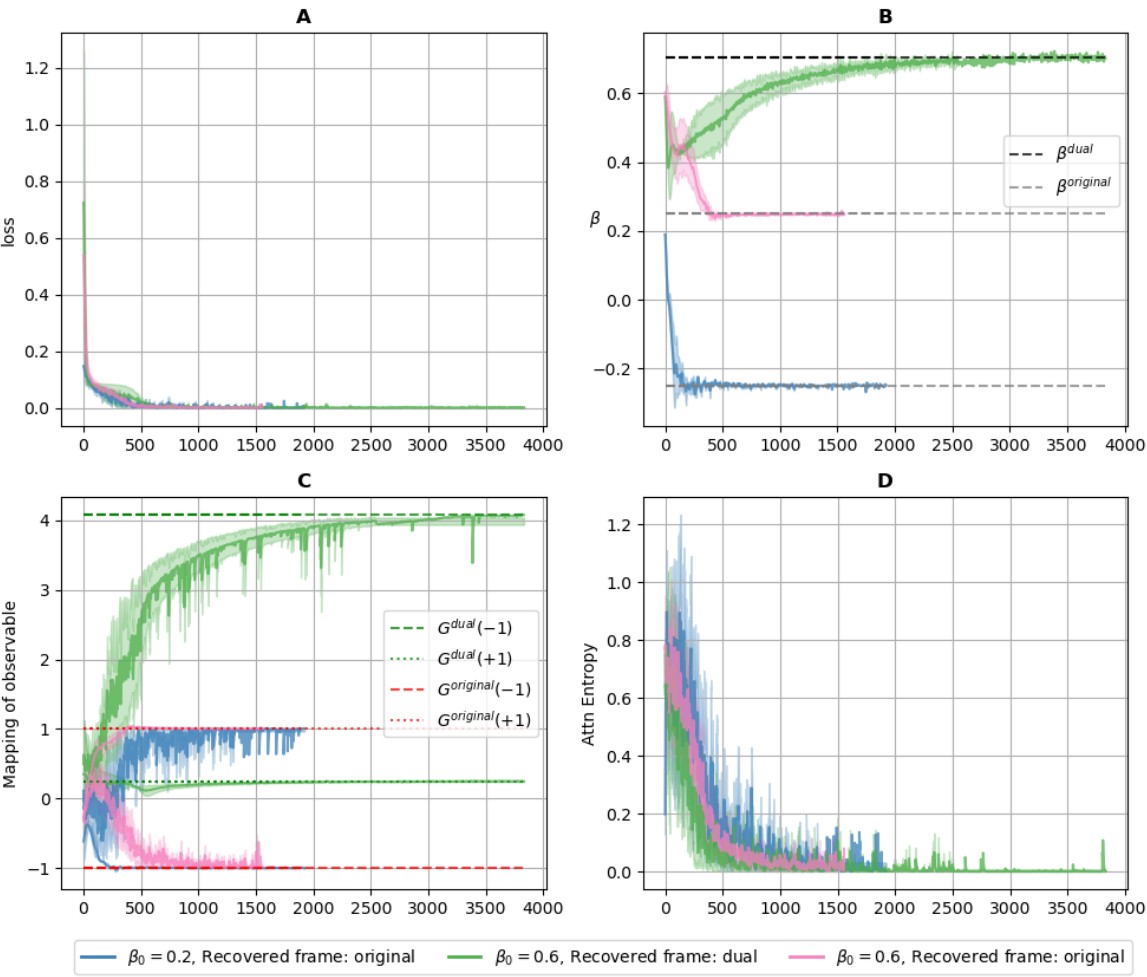

*Figure 11.* Training progress for runs from $\beta = 0.25$, grouped by $\beta_0$ and the final frame discovered to showcase the trajectory of various metrics. We show exponentially smoothed moving average of the following metrics: (A) Loss, (B) $\beta$, (C) Mapping of observables, (D) Entropy of Gumbel-Softmax($\theta_1$) For (B) and (C) we denote theoretically expected values in original and dual frames by the dashed lines. Note that these runs are for recovering the original 2d-Ising model.

## D. Scaling to bigger lattices

Figure 12 shows the fraction of instances in which either $\tilde{\beta}$, $\beta$, or $-\beta$ were successfully recovered. We observe that the convergence rate improves as the lattice size increases to $10 \times 10$, $12 \times 12$, and $14 \times 14$.

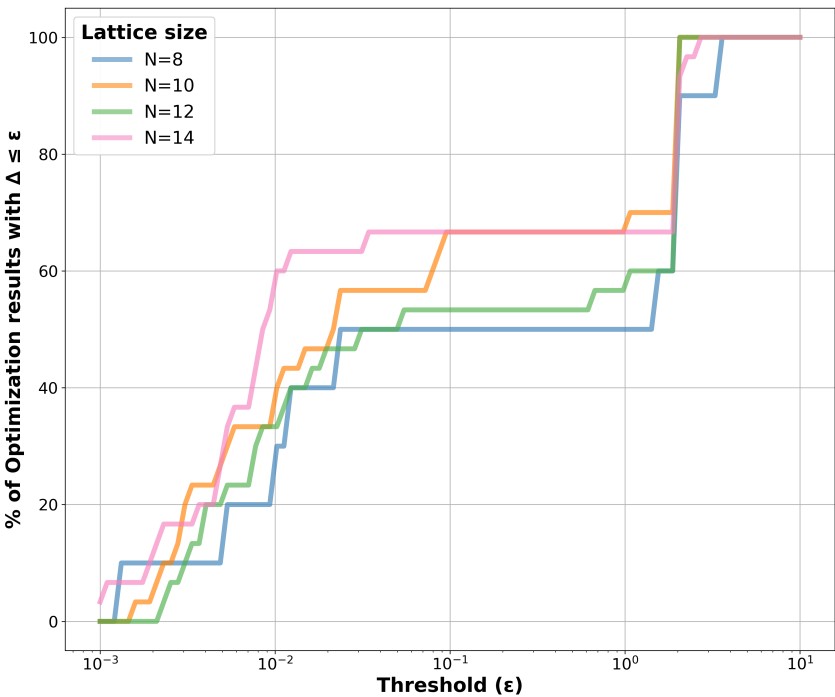

*Figure 12.* Fraction of optimization results with a maximum deviation, $\Delta$ less than threshold, $\epsilon$ for optimization runs on $\beta = 0.25$. We observe that increasing N beyond eight results in only a marginal improvement in performance.

## E. Post-phase transition performance

Figure 13 shows a plot similar to Figure 4 but for the phase transition phase, $\beta > \beta_c$. We observe that the method does not work as well as with the lower $\beta$s.

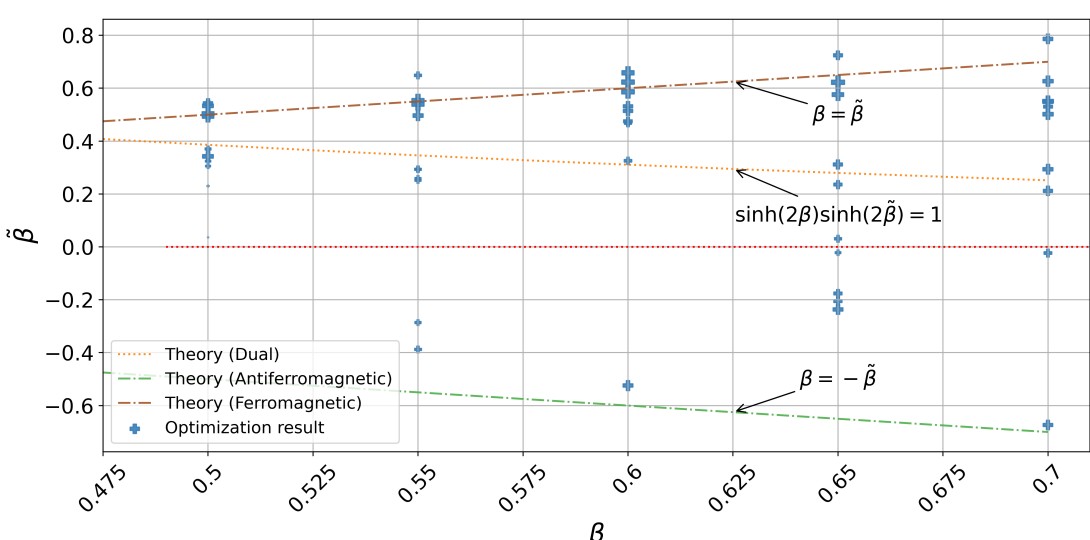

*Figure 13.* Optimization results for post-phase transition don't work as well as before the phase transition.

## F. Empirical support for approximate duals

We now provide evidence to show that a broad set of moments—including correlation length—is accurately matched across approximate duals. This close agreement suggests that the essential physics is preserved; as discussed in the main text we believe this largely follows from the fact that the single-spin-flip probability determines much of the physics in this regime.

To assess generalization, we compare feature statistics between approximate duals (both found from our experiments in the paper and from the hypothesised line $\beta + \kappa = const$) and the corresponding theoretical duals. Importantly,

- We evaluate various features not included in the training loss
- We compute these features on larger lattices of size $24 \times 24$, beyond the training regime
- We include approximate duals along the hypothesised line for comparison

Due to computational constraints, training directly on large lattices like $24 \times 24$ is infeasible. Instead, we apply the learned mappings from $8 \times 8$ lattices to estimate features on larger systems without retraining.

In all the plots, the top panel shows average feature values across all approximate duals for $\beta_0 \in \{0.2, 0.25, 0.3\}$ — the original-frame $\beta$ values that yielded these approximate duals, and the lower panels show individual approximate duals (marked by x). Squares mark the features corresponding to theoretical dual configurations.

We consider three categories of features. For each category, we present two plots: (Framework) one based on approximate duals found by our framework, and (Hypothesized) another based on duals inferred from the hypothesized line $\beta + \kappa = $ const, where the constant is chosen to intersect the known dual point $\beta_{dual}$.

- **Product of consecutive links in a linear chain in a lattice of size 24x24**: There are 24 such features (not used in the training loss). Figure 14 & 15 shows the plots for approximate duals found by the framework and those from the hypothesised approximate duals. Both sets of approximate duals closely match theoretical expectations
- **13 features constructed from link products used in the training loss**: Figure 16 & 17 shows the plots for approximate duals found by the framework and those from the hypothesised approximate duals. These features match well across both sets of approximate duals, despite being trained on smaller 8x8 lattices.
- **101 Features constructed from all possible (up to gauge equivalence) link products in a grid (not used in the training loss)**: Figure 18 & 19 shows the plots for approximate duals found by the framework and those from the hypothesised approximate duals. Even this exhaustive set of features shows strong alignment with the theoretical dual, reinforcing the robustness of our approach.

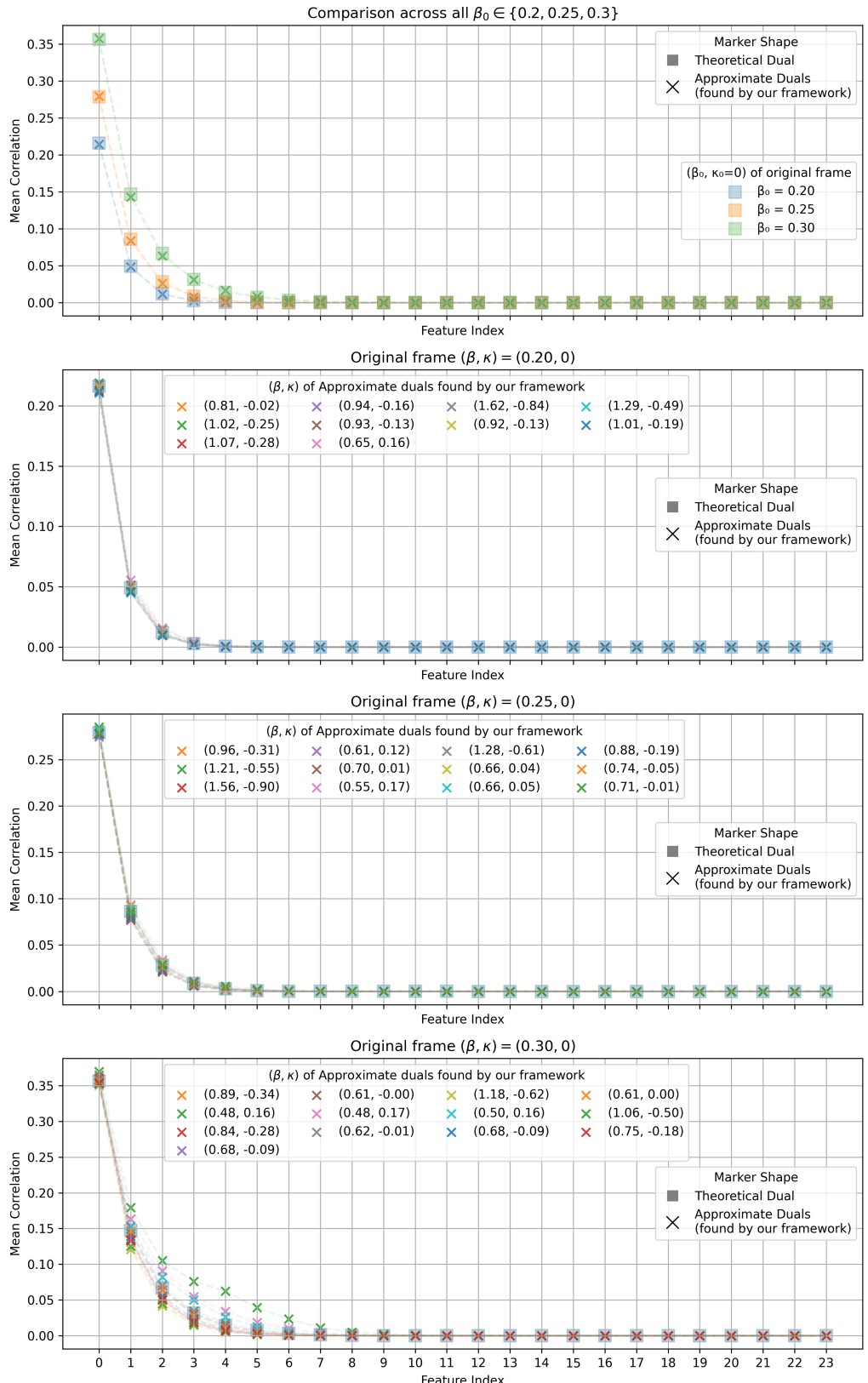

*Figure 14.* (Framework) The moments (product of consecutive links in a linear chain in a lattice of size 24x24) computed from the approximate duals found by our framework closely match those of theoretical duals.

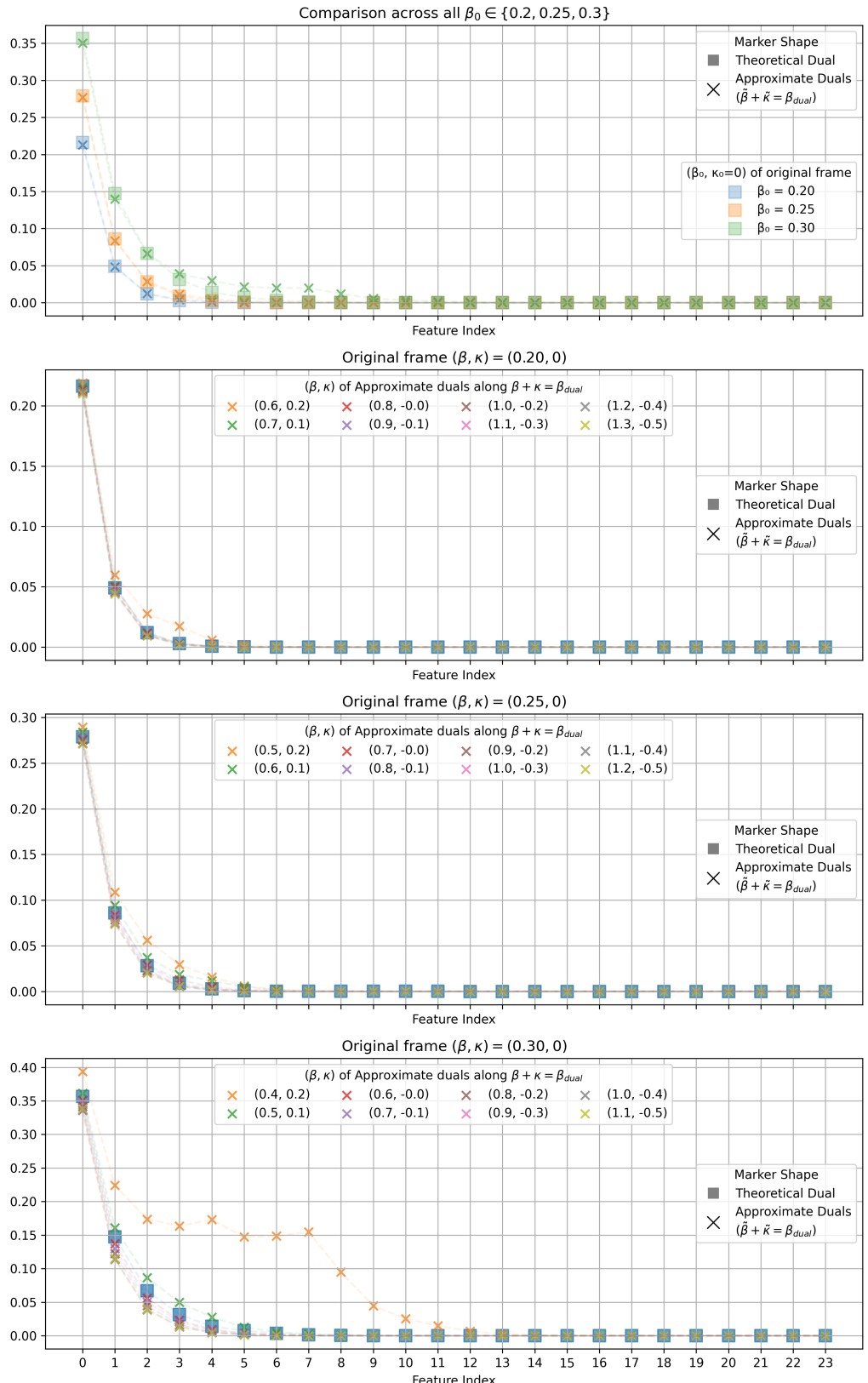

*Figure 15.* (Hypothesis). The moments (product of consecutive links in a linear chain in a lattice of size 24x24) computed from the approximate duals along the hypothesized line ($\beta + \kappa = const$) closely match those of theoretical duals.

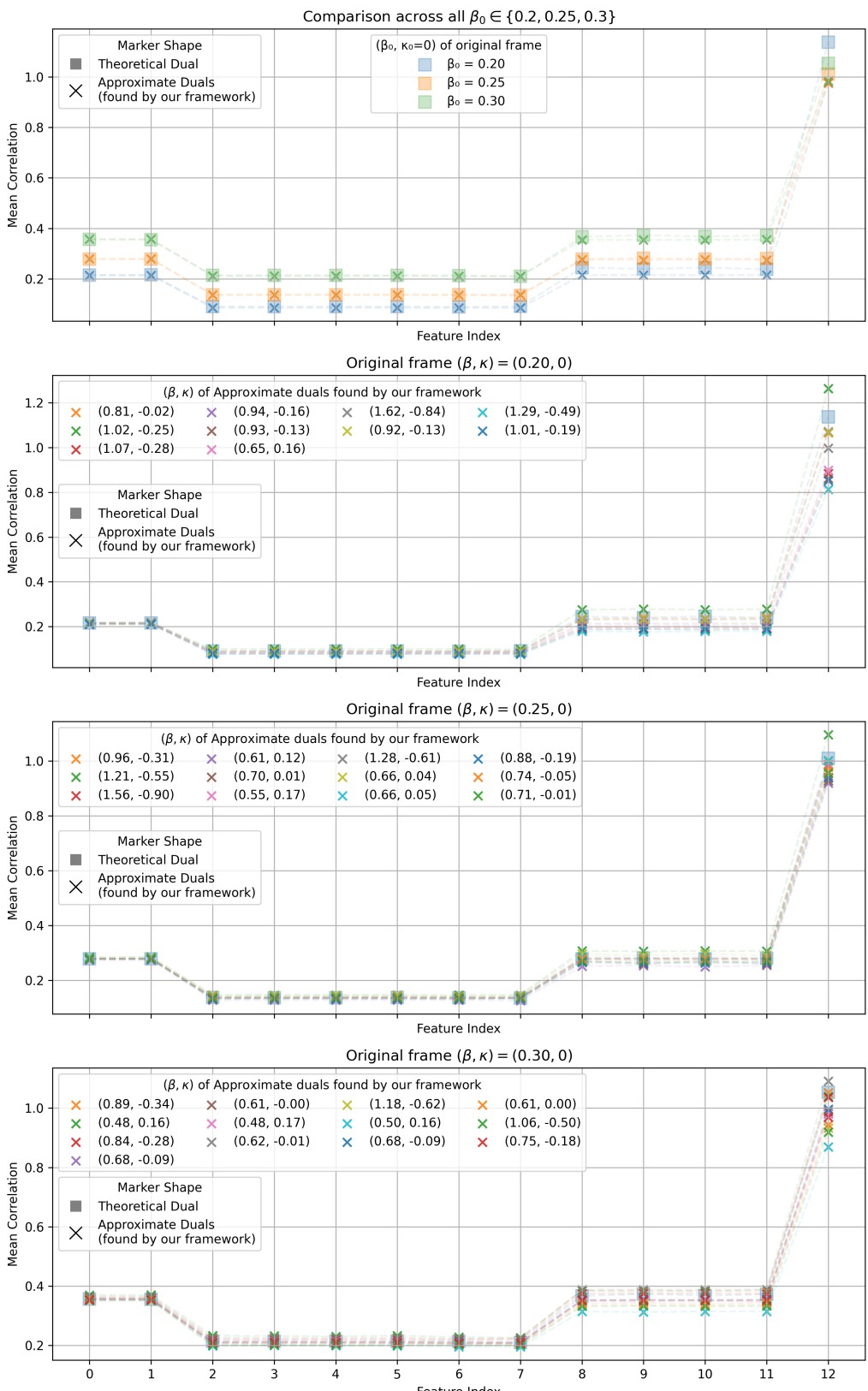

*Figure 16.* (Framework) The moments (13 features constructed from link products used in the training loss) computed from the approximate duals found by our framework closely match those of theoretical duals.

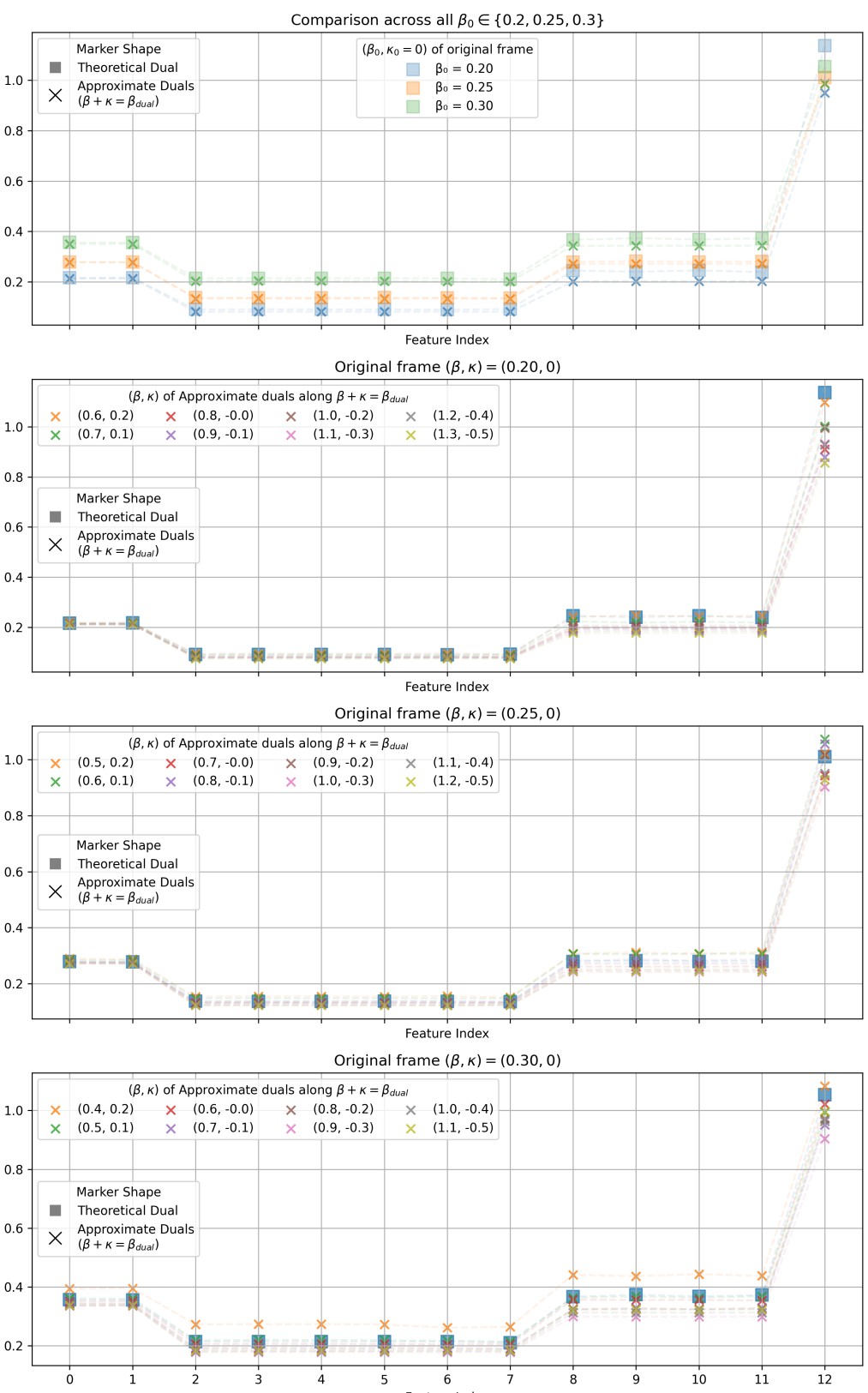

*Figure 17.* (Hypothesis) The moments (13 features constructed from link products used in the training loss) computed from the approximate duals along the hypothesized line ($\beta + \kappa = const$) closely match those of theoretical duals.

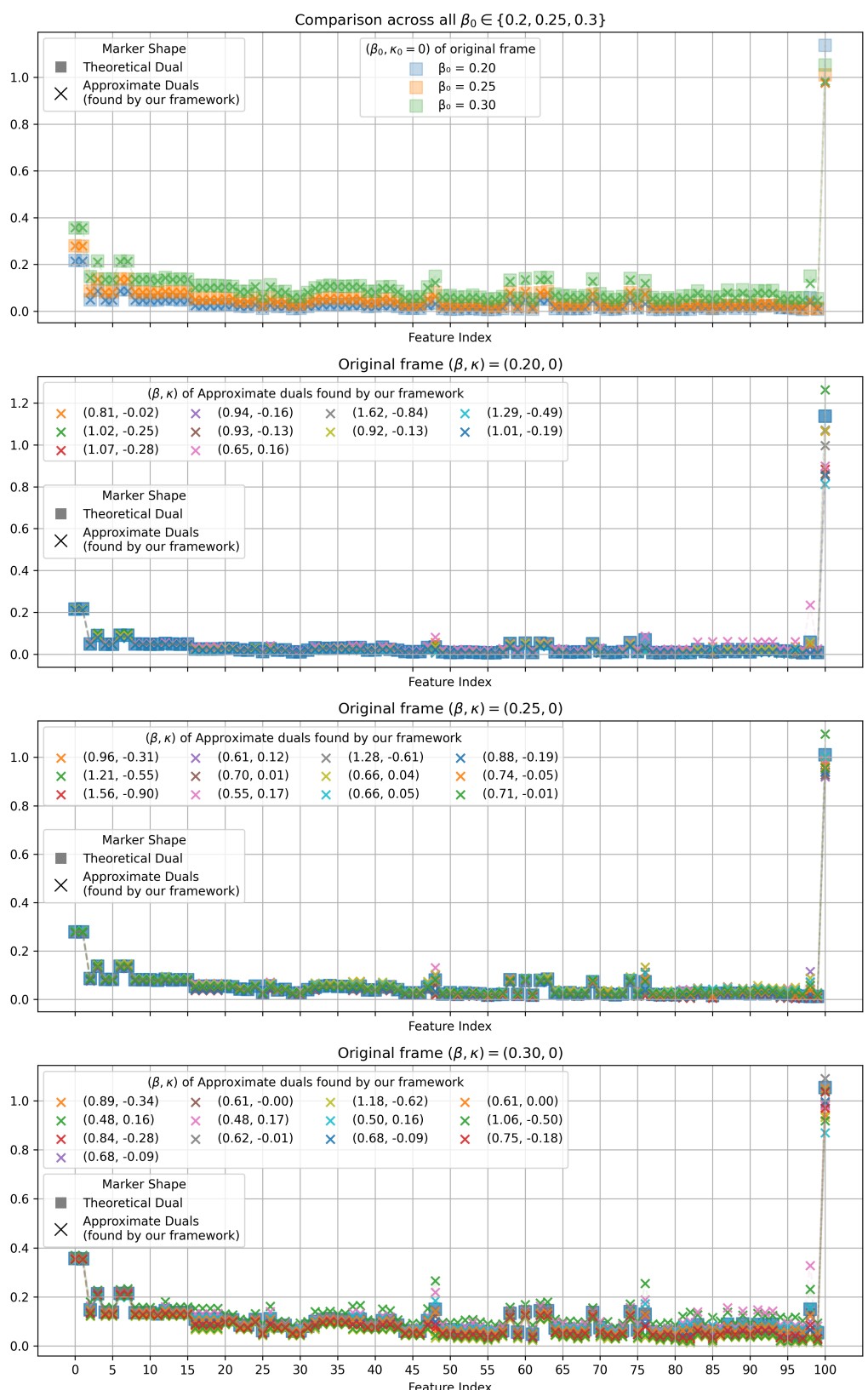

Figure 18. (Framework) The moments (101 Features constructed from all possible (up to gauge equivalence) link products in a grid (not used in the training loss) computed from the approximate duals found by our framework closely match those of theoretical duals.

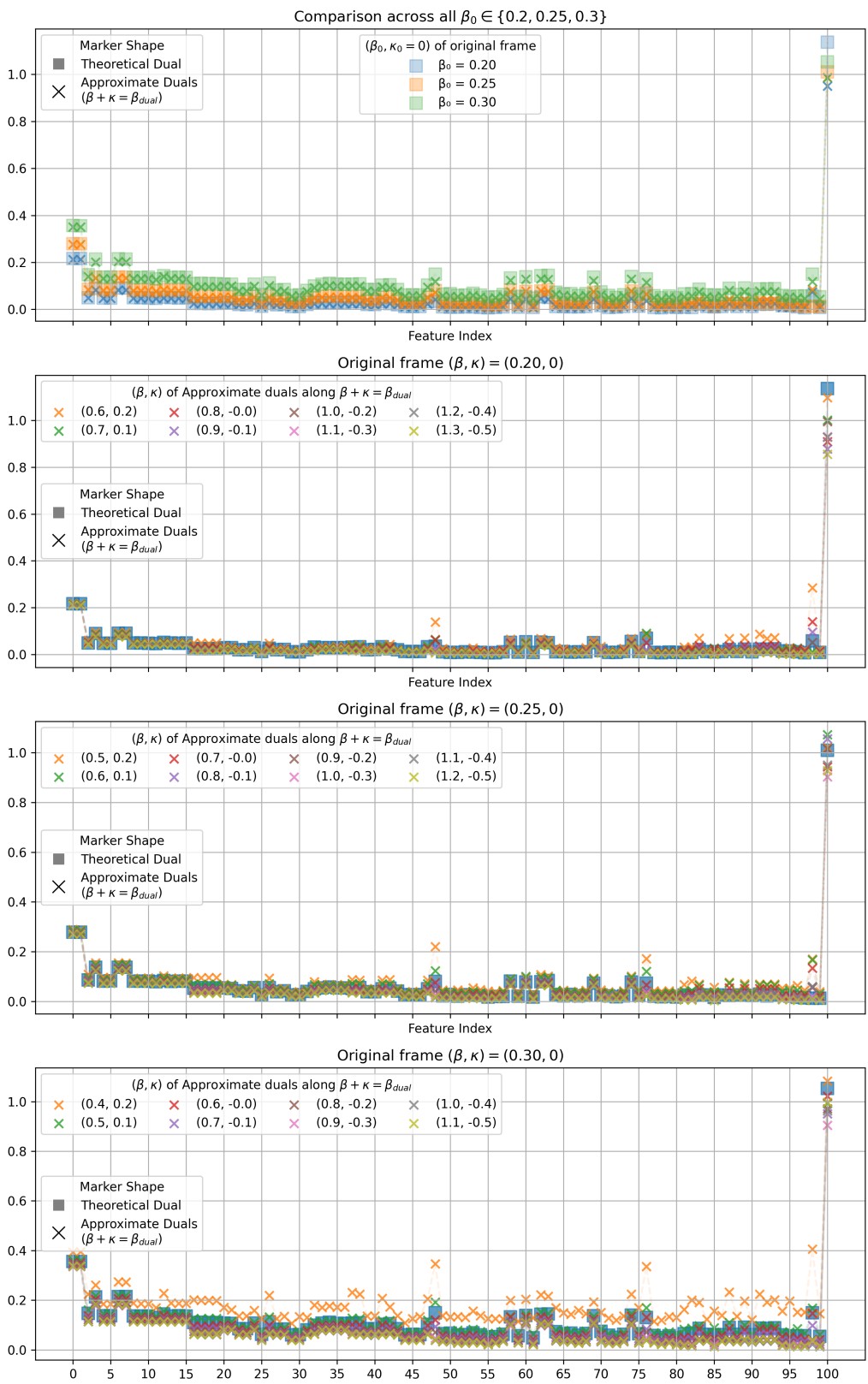

*Figure 19.* (Hypothesis) The moments (101 Features constructed from all possible (up to gauge equivalence) link products in a grid (not used in the training loss) computed from the approximate duals along the hypothesized line ($\beta + \kappa = const$) closely match those of theoretical duals.

