# OpenReview forum: "A Machine Learning Approach to Duality in Statistical Physics"
_ICML.cc/2025/Conference — ICML 2025 poster_

### Official Review · Reviewer_8KMu · 2025-03-10

**Overall Recommendation:** 2

**Summary:**

This paper show cases a machine learning approach to find dual models in statistical physics. The authors outline a training procedure to find an ML model that can match the observables estimated from two statistical physics systems. If such a match is found for two different Hamiltonians then that points to the existence of a duality transform between these two models. The authors discuss a variance reduction technique to improve the convergence of their learning algorithm. They test their method on the 2D Ising model and an Ising model with plaqueete interactions.  They rediscover the duality of the 2D Ising model and draw some interesting conclusions about the plaquette model.

**Claims And Evidence:**

The claims made in the paper are supported by limited evidence. The main claim of the paper is that duality transforms can be discovered by an ML algorithm. This is only demonstrated for the 2D Ising model, which is possibly the simplest example one can demonstrate this on.

**Essential References Not Discussed:**

N/A

**Experimental Designs Or Analyses:**

See questions/suggestions

**Methods And Evaluation Criteria:**

Yes they do make sense. But a larger suite of experiments would have been ideal

**Other Comments Or Suggestions:**

The paper can be improved by:

> More experiments that show that this works in non-trivial settings.
> Better figure captions and may be even a different approach to presenting the results visually. For instance, figure 6 is very hard to parse. It would be better if it is split up into smaller figures that support specific conclusions in the text.
> Overall this is a good idea but might be better suited for a physics journal. The ML approaches used here are fairly simple and they do not have wide scale applicability beyond studying stat. phys. models.

**Other Strengths And Weaknesses:**

Strengths:

This is a well written paper which deals with a very interesting problem in statistical physics. The results are very encouraging and the authors are very straight forward with their findings and future directions.

Weaknesses:
The main weakness of this work is limited experiments. The only experiment that shows that such a duality transform can be learned is done for the 2D Ising model. This is the simplest case with a known answer. For the plaquette model, the method seems to find some sort of approximate duality in the ordered phase, but the evidence for this is weak. Also, there must be a  set "testing" experiments that show that the learned dual model reproduces the essential physics of the original model.

**Questions For Authors:**

> For Fig.6 the learned points actually seem to deviate from the \beta  + \kappa line. What is the cause of this error?

**Relation To Broader Scientific Literature:**

Duality is an important notion that used to study models in statistical physics. The notions of duality used in this paper are derived from existing works in the filed (eg. https://journals.aps.org/rmp/pdf/10.1103/RevModPhys.52.453)  I am not aware of papers that try to learn duality transformations explicitly from data.

**Theoretical Claims:**

There are no major theoretical claims here to check.

---

> ### Author Rebuttal · Authors · 2025-03-31
>
> ## **Stronger evidence of approximate duality**
>
> We thank the reviewer for the suggestion and agree that additional evidence strengthens our claims. We now provide such evidence, showing that a broad set of moments—including correlation length—is accurately matched across approximate duals. This close agreement suggests that the essential physics is preserved, a novel result to our knowledge.
>
> To assess generalization, we compare feature statistics between approximate duals (both found from our experiments in the paper and from the hypothesized line $\beta + \kappa = const$) and the corresponding theoretic duals. Importantly,
> - We evaluate various features not included in the training loss
> - We compute these features on larger $24 \times 24$ lattices, beyond the training regime
> - We include approximate duals along the hypothesized line for comparison
>
> Due to computational constraints, training directly on large lattices like $24 \times 24$ is infeasible. Instead, we apply the learned mappings $G_{\theta}$ from $8\times8$ lattices to estimate features on larger systems without retraining.
>
> In all the plots, the top panel shows average feature values across all approximate duals for $\beta_0 \in \\{0.2, 0.25, 0.3\\}$ —the original-frame $\beta$ values that yielded these approximate duals, and the lower panels show individual approximate duals (marked by x). Squares mark the features corresponding to theoretical dual configurations.
>
> We consider three categories of features. For each category, we present two plots: (Framework) one based on approximate duals found by our framework, and (Hypothesized) another based on duals inferred from the hypothesized line $\beta + \kappa = const$, where the constant is chosen to intersect the known dual point $\beta_{\text{dual}}$. All links point to an anonymous repository hosting the plots.
> - **Product of consecutive links in a linear chain in a lattice of size 24x24:** There are  24 such features. (not used in the training loss)
>   - Framework (https://bit.ly/4hQnMJH)
>   - Hyopthesized (https://bit.ly/4jhTOj7)
>   - Both sets of approximate duals closely match theoretical expectations
>
> - **13 features constructed from link products used in the training loss:**
>    - Framework (https://bit.ly/4lq2vtw)
>    - Hyopthesized (https://bit.ly/41TzC16)
>    - These features match well across both sets of approximate duals, despite being trained on smaller 8x8 lattices.
>
> - **101 Features constructed from all possible (up to gauge equivalence) link products in a $3\times3$ grid**:   (not used in the training loss)
>    - Framework (https://bit.ly/4lt5Tnx)
>    - Hyopthesized (https://bit.ly/42rEtGM)
>    - Even this exhaustive set of features shows strong alignment with the theoretical dual, reinforcing the robustness of our approach.
>
> Regarding Figure 6, we appreciate the suggestion and will revise the caption to provide a clearer explanation, helping readers better interpret the figure.
>
> ## **Simplicity of ML methods**
>
> Although the ML methods are indeed simple, they certainly involve new important ideas; we did not find anywhere in the literature the task of learning both parameters in a Hamiltonian and an observable. We also thought that the application of these methods to statistical physics models was sufficiently broad, since it is one of the major branches of theoretical physics.
>
> We could however have further emphasised the broader applicability of these methods; one such is Hamiltonian truncation, where the task is to compute the spectrum of some truncated hamiltonian to gain insights into the behaviour of the physical systems; it is routinely applied to the study of QFT phenomena, such as duality. RG flows, etc. Our methods could for instance be used in this context to discover accidental dualities like Chang duality (https://journals.aps.org/prd/abstract/10.1103/PhysRevD.93.065014), which have so far received no systematic explanation. More precisely, the task in this context would be to fit parameters in a truncated hamiltonian such that the distance between the eigenvalues of the hamiltonian, or the eigenvalues of some related observable of different models are minimised. This is precisely the type of task discussed in this paper.
>
> ## **Deviation from the hypothesized $\beta + \kappa$ line**
>
> We believe this happens because the idea that the physics is completely determined by $\beta + \kappa$ (as explained on p6, where this determines the single spin-flip probability) is an approximation; in practice there is always a finite probability for two spins to flip together, and in this case the observables will depend on \beta and \kappa individually. We believe this is the reason for systematic movement off of the $\beta + \kappa$ line.
>
> We stress there is also likely a statistical reason for the movement off the line; not all runs find equally good minima of the loss due to random initializations etc. There is scope for improvement of this through engineering improvements.

---

### Official Review · Reviewer_aEBk · 2025-03-13

**Overall Recommendation:** 3

**Summary:**

A methodology is developed for automatically finding duality transformations
on  lattice gauge theory models (actually simple 2D Ising models are considered
in this paper as a proof of concept). This duality transformation is an important tool in physics as it
can allows one to access physical properties of systems in the so-called non-perturbative regime, i.e.
when the coupling constant are too strong to perform any perturbative expansions analysis which is otherwise intractable.

**Claims And Evidence:**

The paper constitutes mainly a methodological proof of concept based on experimental observations.
It is shown for instance that on the 2d Ising model, dual couplings and dual variables on the dual lattice  are
properly recovered numerically over some range of temperature.

**Essential References Not Discussed:**

I did not find that an important reference was missing.

**Experimental Designs Or Analyses:**

The NN parameterizing the dual model is a simple Boolean vector function using Gumbel soft-max to enforce sparse solution for the weights, taking as input as set of candidate binary links values (+-1) and returning a moment value.
The set of compound moments which are used to define the loss are not given, neither the code. I believe the experiments are correctly done, but these are not easy to reproduce with the information provided in the paper.

**Methods And Evaluation Criteria:**

The method is based on moment matching with MCMC sampling and a property of duality that relates primal and dual moments, involving nearest neighbour variables. A certain number of well (manually) chosen link product of these variables
are estimated by sampling both the primal model and a candidate dual model parameterized by a neural network. The optimization stops when primal and candidate dual moments coincide. The methodology makes sense for the problem  at hand, and gives in principle
more precise information than previous methods cited in the paper.

**Other Comments Or Suggestions:**

no additional comments

**Other Strengths And Weaknesses:**

The proposed method is sound, the interpretation of the results shown is quite clear but the experimental test is very limited.
The question addressed in this work is interesting but it looks to me as a  preliminary work as the method is tested on elementary example (the 2D Ising one), and a less elementary one, namely 2D Ising + plaquette four spin interactions. But on this last example no duality transformation is to be found in general and leads experiments to output kind of spurious results . More generally I am a bit perplex about the overall practical interest of this study, which appears to me as being mostly at the level of a  curiosity without practical use.
Usually duality transformation are used to address the strong coupling regime in order be able to compute n-points correlations.

**Questions For Authors:**

I would like to ask more on the general scope of the paper.
Most of,  if not all (would you agree? ) duality transform correspond to a Fourier transform over some compound primal degrees of freedom (see e.g. Cardy, "Scaling and Renormalization in Statistical Physics"). Do the authors expect to unveil new kind of dual transformations with this method? Ultimately what kind of models would that be of interest to learn new physics? is it realistic considering the fact that the method requires to sample at least once a model in the strong coupling regime?

In complement to this, here the method requires to sample both the primal and all candidate dual models, so  that I would expect that the relevant setting corresponds to the primal being strongly coupled, so that we can hope to be able to sample easily all the dual candidates. Then How is it that the most convincing experiment displayed corresponds the (wrong?) setting, i.e. when the primal model is  weakly coupled (high temperature).

**Relation To Broader Scientific Literature:**

There are few references on the subject, the main one seems to be Betzler,Krippendorf  which basically motivates the problem and propose unsupervised methods to find duality relations. It is cited but not much discussed, in particular comparative  merits with respect to the overall motivation is not detailed.

**Theoretical Claims:**

There are no theoretical claims.

---

> ### Author Rebuttal · Authors · 2025-03-31
>
> ## **Code Availability & Moments & Reproducibility**
>  We would like to kindly remind the referee that a full working code was provided in the supplementary material. Feature computation is handled by `src.utils_ising.generate_masks` (which generates 13 link product masks) and `src.utils_ising.feature_samplewise`. We will also open-source the code and experimentation scripts to ensure full reproducibility.
>
> **Training Features**
> We use a specific set of link product moments in the training loss. These are visually illustrated in this plot (https://anonymous.4open.science/r/temp_rebuttal_repository-0225/images/features.png), where the red highlights indicate the link products averaged across the lattice and samples to form statistical features.
>
> ## **How is our work different from  Betzler,Krippendorf ?**
>
> Betzler-Krippendorf has a somewhat different focus, which includes an investigation of the advantage of a variety of different known dual representations. The part of this paper with the most overlap with ours is Section 3.3. The key differences are the following: (1) the input into the duality mapping is spin configurations sampled in the original frame, after which a *second* step of sampling is done: this is not the usual setup for duality in physics, where one usually just samples once (importantly, in a dual frame) and then performs a deterministic mapping, as in our work. (2) It seems that the loss function used in that work cannot be formulated unless the duality mapping of temperatures is known already, and thus that this work cannot be used to find *new* dualities, which our formalism in principle allows.
> We will include this discussion in the updated version of our manuscript.
>
>
> ## **Practical use case of our work**
>
> We have a somewhat broader perspective on duality transformations: we do not think that they are used only to compute n-point functions. Rather they provide an important perspective on the structure of the underlying physics. In particular, Kramers-Wannier duality is important not for calculations because it is the simplest example of an order/disorder duality. The discovery of a genuinely new duality – based on novel principles – would have a huge impact on physics in general.
>
> In addition to the above orientational remarks, we would like to provide two more concrete directions that we hope to explore with this technology on a short time scale.
>
> 1. Much recent work (summarized in https://arxiv.org/abs/2308.00747) has focused on points in parameter space which are *self-dual*, in which there is a new symmetry of an exotic “non-invertible” type. Little is known on how these symmetries work in more general theories (such as the plaquette Ising model) – for example we do not know whether the symmetry exists on the phase-transition line away from \kappa = 0. Understanding whether or not a self-duality can exist at all away from \kappa = 0 is an important first step in understanding the symmetry.
>
> 2. There exist models with interesting phase transitions (e.g. the simple case of bond percolation, reviewed in https://arxiv.org/abs/math-ph/0103018) that are not described by any *known* local model. It is very interesting to ask whether there can be a (perhaps complicated) local model that describes them.
>
> In both of these cases, we anticipate learning about deeper questions from the existence of a duality. In our eyes this is the true value of our work.
>
> ## **Do Dualities Reduce to Fourier Transforms?**
>
> In our opinion, many dualities go beyond Fourier transformations (a well-known example is AdS/CFT, which has no simple Fourier interpretation at all). As described above, our hope indeed is that other kinds of dualities – based on fundamentally new ideas – could be unveiled using these methods. One concrete example is Chang duality in \phi^4 theory (https://journals.aps.org/prd/abstract/10.1103/PhysRevD.93.065014). As we comment in the rebuttal to referee 3, our methods could realistically be applied to the discovery of such dualities, which are under intense investigation in the hep-th community.
>
> Let us also note that our approximate dualities are already quite interesting and novel!
>
>
> ## **Sampling in the strong coupling regime**
>
> For these models the difficulty of sampling using MCMC in both the strong and weakly coupled regime is basically the same. (Difficulties arise close to the critical point where critical-slowing down causes familiar but surmountable problems). Obtaining analytic results (e.g. Feynman diagrams etc.) is of course hard at strong coupling, but – very importantly – our formalism does not require this at all.
>
> ## **Phase Dependence of the Method’s Performance**
>
> For statistical physics model this strong-weak paradigm is not really relevant; we should stress that it is equally easy to sample in both phases. The reasons why our methods work best in one phase seem more subtle and related to well-known difficulties encountered in the inverse Ising problem.

---

### Official Review · Reviewer_k86m · 2025-03-14

**Overall Recommendation:** 3

**Summary:**

This paper develops a machine learning method to find dualities in statistical physics models. The authors turn duality discovery into an optimization problem by using neural networks to map between original and dual models. They create a loss function that matches correlation functions between the two descriptions. The method successfully finds the known Kramers-Wannier duality for the 2D Ising model, getting both the temperature mapping and how observables transform. They also study the plaquette Ising model, showing it likely doesn't have simple self-dualities, though they find interesting "approximate dualities" in ordered phases. The work carefully analyzes optimization problems and uses variance reduction to make training work better.

**Claims And Evidence:**

The main claims are supported by numerical results. The rediscovery of Kramers-Wannier duality seems convincing. Their negative result about the plaquette model's self-dualities is also supported by extensive testing. The explanation of why certain parameter combinations give similar results in ordered phases seems insightful.

**Essential References Not Discussed:**

The reference seems comprehensive. Additional discussions on neural networks with symmetries seem relevant.

**Experimental Designs Or Analyses:**

The experiments reveal real some physical insights. The method struggles near the critical point and in the ordered phase, which can reflect actual physics rather than just technical problems. Their analysis of how the attention picks out the dual lattice is reasonable. The tests of different lattice sizes show what's practical while being honest about computational limits.

**Methods And Evaluation Criteria:**

The methods make physical sense. Using neural networks to map observables gives enough flexibility while keeping physical meaning. Their loss function based on correlations is reasonable for comparing physical systems.

**Other Comments Or Suggestions:**

No other comments.

**Other Strengths And Weaknesses:**

The paper's main strength is its reframing of duality discovery as machine learning. The physics interpretations are clear and they're honest about the limitations. The main drawbacks are computational speed and handling more complex models. Additional discussions on what the approximate dualities tell us about ordered phases can be beneficial.

**Questions For Authors:**

1. How would the method work for systems with continuous symmetries where there are many more possible dualities?
2. Is it possible to adapt this to quantum systems by working with transfer matrices?
3. Is it worth exploring other neural networks that might better capture long-range correlations?

**Relation To Broader Scientific Literature:**

This work combines statistical physics and machine learning in a novel way. While others have used machine learning for physics problems, turning duality discovery into optimization is novel. It adds to existing analytical methods for finding dualities while potentially letting us find new ones.

**Theoretical Claims:**

The theoretical claims are in in general justified.

---

> ### Author Rebuttal · Authors · 2025-03-31
>
> We thank the referee for their careful reading and interesting questions. We believe each of their three questions are interesting starting points for future research, and our more precise responses are below:
>
> 1. **How would the method work for systems with continuous symmetries where there are many more possible dualities?** In principle, the philosophy of the method – matching moments of an appropriate subset of observables – would still work for a system with continuous symmetries (e.g. we see no real reason why we could not obtain particle-vortex duality in 3d systems with a continuous U(1) symmetry, etc.). In practice, it seems conceivable that computationally this would be more onerous, as it would take longer to equilibrate the system when doing MCMC sampling (generally there would be gapless modes associated with Goldstone modes of the continuous symmetry which take longer to thermalize).
>
> 2. **Is it possible to adapt this to quantum systems by working with transfer matrices?** We think that it should be possible to use a similar approach for quantum systems without a sign problem: we would use the usual Trotter approach to reduce the quantum problem to an effective classical problem that could be simulated using very similar techniques. The problem would now be strongly anisotropic (as the time direction is now quite different from the spatial ones) and some thought should be put into the correct form of the neural network that maps observables (presumably it would also be strongly anisotropic) but in our opinion this would be manageable.
>
> 3. **Is it worth exploring other neural networks that might better capture long-range correlations?** Yes: making the neural network for the mapping have a larger spatial extent (i.e. going to “n” hops away on the lattice where n is a small number) is something that we are currently actively investigating as indeed some systems might require the need to capture longer-range correlations. Training becomes more difficult as this mapping is made more and more non-local. In principle there are known examples of duality (AdS/CFT being a particularly dramatic example) where the mapping is completely non-local; we do not think that our approach will reliably be able to find such examples as the space of mappings is simply too large.
>
> ---
> **Finally, in response to the question about what the approximate duals reveal about the ordered phase:**
>
> The approximate duals essentially describe a regime where the physics can be approximated by isolated single-spin flip events; this explanation shows that many different models are expected to flow to this regime, giving an example of the universality of RG (and conversely highlighting the difficulties in pinpointing the original Hamiltonian from data in such a regime)

---

> > ### Comment · Reviewer_k86m · 2025-04-04
> >
> > Thanks for the detailed rebuttal. I would like to keep my score.

---

### Decision · Program_Chairs · 2025-05-01

**Decision:**

Accept (poster)

**Comment:**

As noted by the referees, the submission's strenght is its novelty, namely the proposed mapping of a genuine, fundamental statistical physics problem, i.e., searching for duality, in a machine learning optimization problem. The approach appears to have natural practical limitations and in particular might be computationally challenging in complex models: as stated by the referees, the investigation can be seen more as a "proof of concept" on the simplest possible model of a general strategy that might otherwise be hard to apply. However, as first on the proposed method, the paper can be nevertheless inspiring for future investigations in statistical physics, as the theoretical investigation can highly benefit from the discovery of fundamental properties such as duality. I recommend the authors to expand the comparison of their work with the one of [Betzler and Krippendorf](https://onlinelibrary.wiley.com/doi/abs/10.1002/prop.202000022). In light of the overall positive feedback and the novelty, I recommend *Weak Accept*.